# Laryngeal Papillomatosis

**DOI:** 10.3390/cancers17060929

**Published:** 2025-03-10

**Authors:** Jared Levinson, William Edward Karle

**Affiliations:** Department of Otolaryngology, Northwell, New Hyde Park, NY 11042, USA; jlevinson1@northwell.edu

**Keywords:** laryngeal papillomatosis, recurrent respiratory papillomatosis, human papillomavirus, airway obstruction

## Abstract

Laryngeal papillomatosis is the most common benign disease of the larynx that affects both children and adults. It is caused by the human papillomavirus. While it is a benign disease, it can cause significant problems with the voice and breathing. The disease tends to be more aggressive when it presents in childhood and is difficult to manage because of its tendency to recur. The most common treatment is surgery, with other additional treatments available. This review aims to provide an overview of laryngeal papillomatosis and the various surgical and medical management options that are currently available or under investigation.

## 1. Introduction

Recurrent respiratory papillomatosis (RRP), also referred to as laryngeal papillomatosis (LP) when primarily involving the larynx, is the most common benign neoplasm of the larynx [1]. It is caused by the human papilloma virus (HPV), predominantly subtypes 6 and 11 [2,3]. While RRP is a benign disease, it remains challenging to treat with significant morbidity due to its high recurrence rates [1]. The incidence in the United States has previously been estimated to be 4.3 per 100,000 children and 1.8 per 100,000 adults but has recently been shown to be declining as a consequence of the creation and subsequent expansion of HPV vaccination [1].

RRP exhibits a bimodal distribution classified as juvenile-onset (J-RRP) and adult-onset (A-RRP). Compared with A-RRP, J-RRP is typically characterized as a more aggressive disease with higher rates of recurrence, often requiring frequent surgical interventions. J-RRP is thought to arise from vertical transmission during childbirth from infected genital mucosa, while A-RRP arises from sexual transmission [1].

## 2. HPV

To understand this disease, it is helpful to first appreciate the infective vector which causes it. HPV comprises a family of small double-stranded DNA viruses that infect the cutaneous and mucosal epithelium. Over 150 HPV types have been identified [4]. HPV types that infect the cutaneous epithelium cause common skin warts, while HPV types that infect the mucosal epithelium are further categorized as ‘high risk’ or ‘low risk’ based on their oncogenic potential [4].

High-risk HPV types are implicated in the development of cervical cancer, as well as oropharyngeal and anogenital cancer. The high-risk types include HPV 16, 18, 31, 33, 45, 52, and 58, with HPV 16 and 18 accounting for about 66% of cervical cancers [5]. Low-risk HPV types include HPV 6 and 11, which more frequently cause low-grade genital lesions, anogenital warts, and RRP [5]. HPV infection occurs at the basal lamina of the epithelium, and most infections are asymptomatic without any clinical manifestations. Despite the high incidence of infection, the majority resolve spontaneously in 1–2 years [5]. Those who develop persistent infections are most at risk for developing HPV-related cervical, anogenital, or oropharyngeal cancers [4,5,6].

HPV is the most common sexually transmitted infection in both the United States and throughout the world. Prior to the introduction of the HPV vaccine, 79 million people were infected in the United States, with about 14 million new infections per year. Almost half of these infections were in people aged 15 to 24 years old [6].

In 2006, the quadrivalent HPV vaccine was introduced in the United States. Data throughout 2018 revealed an 86% decrease in the prevalence of HPV types 6, 11, 16, and 18 among females 14 to 19 years old, and a 71% decrease in females 20 to 24 years old. A nine-valent HPV vaccine was introduced later in 2014 [7,8]. In the head and neck, 80% of oropharyngeal squamous-cell carcinoma (OPSCC) is associated with HPV in the US [9]. Between 1988 and 2004, HPV-related OPSCC increased by 225%, while HPV-negative cancers declined by 50% [10].

## 3. Laryngeal Papillomatosis

The most dreaded manifestation of HPV infection within the head and neck region is cancer, but its benign presentation as papillomas can have significant consequences as well. Within the larynx, the clinical presentation depends on the extent and location of the papillomatous growths. In children, the characteristic triad of symptoms includes dysphonia, stridor, and dyspnea [11]. J-RRP represents a distinct form of the disease in comparison with A-RRP. LP is typically more aggressive in children, more expensive to treat, and exhibits higher recurrence rates compared with adults [12,13]. Specifically, children with aggressive disease require more frequent procedures and may be more likely to present with more distal disease, require tracheostomy, and experience complications [13,14,15,16,17,18]. It is important to note that adults can still present with aggressive disease. While there is no formal definition of what constitutes “aggressive” disease, a widely published definition includes one or more of the following: 10 or more total surgeries, 4 or more surgeries per year, distal spread to the trachea or beyond, or a need for tracheostomy at any point [13,14,15,16,17,18].

HPV 11 is typically thought to be associated with a greater risk of developing RRP and more severe presentations, which has been widely published [13,15,16,19,20,21,22]. However, there is evidence that age of onset is a more relevant factor than the HPV subtype [12,18]. When controlling for age, the HPV subtype was not strongly associated with disease severity, and the findings were more suggestive that age at diagnosis was more predictive of disease severity. Of patients who were diagnosed between the ages of 0 and 5 years, 80% experienced severe disease. In comparison, 60% of patients diagnosed between the ages of 5 and 10 years had severe disease, and 30% diagnosed after the age of 10 years had severe disease. The authors identified 5 years of age to be a critical age of diagnosis in predicting a severe disease course [18].

It is unclear why children are more likely to present with aggressive disease. While J-RRP is strongly associated with perinatal HPV exposure, few infants who are exposed to or infected with HPV go on to acquire RRP [23,24]. It is also currently unclear why some patients acquire RRP and others do not. Studies have explored the roles of genetic, immunologic, and degree of exposure in the development of RRP [22,25].

The critical period for the development of the immune system in children is ages 4–6 years old (Kovalenko). This coincides with the previously reported critical age at diagnosis for RRP that is associated with more severe disease [26]. Given HPV’s opportunistic behavior (as evidenced by increased HPV-related disease in immunodeficiencies), an immature immune system may explain why younger age at diagnosis has been associated with more severe disease [22]. Regarding genetic factors, certain HLA class II genes are upregulated in RRP, while specific haplotypes are associated with increased susceptibility to severe disease [22].

Cases of malignant transformation are rare but have been reported [27,28,29,30]. Cases of malignant transformation in children were more likely to have pulmonary involvement and a younger age of onset. In adults, dysplasia or malignant transformation was associated with older age of onset. Gender, smoking history, number of operations, and use of cidofovir were not associated with the development of dysplasia or malignant transformation [30]. HPV type has not been shown to be correlated with malignant transformation [31]. The only significant risk factor for malignant transformation is LP without demonstrable HPV DNA, and, thus, HPV typing should strongly be considered for laryngeal disease [31].

Transmission of HPV in children has been suggested to occur via three mechanisms: vertical transmission between a mother and newborn at birth via contact with the genital mucosa, vertical transmission in utero, and horizontal transmission via the child’s environment [22]. A large case–control study of 3033 Danish infants showed that the risk of J-RRP was 231.4 times higher in children whose mothers had genital warts during pregnancy compared with those whose mothers did not. This study also showed that the risk of RRP was doubled in spontaneous vaginal delivery lasting more than 10 h [32].

While cesarian section has been shown to reduce the transmission of viruses such as herpes simplex, hepatitis B, and HIV, it is unclear if there is any role in preventing vertical transmission of HPV [33]. Vertical transmission may still occur in utero or postpartum even in cases of cesarian section and can occur from sperm during fertilization [25,34,35]. Horizontal transmission may occur shortly after delivery through close physical contact with caregivers, relatives, and other environmental exposures that can transmit infection [22].

## 4. Operative Management

Once diagnosed, the mainstay of treatment is the surgical removal of papillomas. Eradication of disease remains difficult due to the propensity of these papillomas to recur [36]. There are a variety of surgical modalities for ablating papillomas, with the most widely used methods including laser, microdebrider, and coblation. Comparative studies have found no significant difference in recurrence trends between these modalities [37].

The carbon dioxide (CO_2_) laser is commonly used for laryngological procedures. The wavelength of the CO_2_ laser is 10,600 nm and is absorbed preferentially by water in soft tissues. The CO_2_ laser can be utilized to cut or ablate tissue [38]. It is imperative to adhere to proper laser safety protocols as airway fire is one of the most serious risks when using a laser. Different power settings and firing modes can be employed for a range of functions. A power of 4–8 W with non-continuous firing allows for more precision when working at the vocal fold level and has a lower level of collateral damage [38].

The potassium-titanyl-phosphate (KTP) laser and TruBlue (A.R.C. Laser GmbH) laser are both considered photoangiolytic lasers that are absorbed preferentially by oxyhemoglobin. The KTP laser has a wavelength of 532 nm, while the TruBlue laser has a wavelength of 445 nm [39]. These lasers are considered to provide more selective ablation of highly vascularized tissues. The angiolytic nature of these lasers allows them to achieve better hemostatic control compared with other lasers [38,40,41]. However, these lasers can only be used as a fiber as opposed to the CO_2_ laser, which has the option of either a fiber or the use of a beam with a micromanipulator. A lower energy setting with an extended pulse width can be used to effectively coagulate papillomatous lesions and preserve the superficial lamina propria by reducing thermal injury [42].

Powered laryngeal shavers have been shown to be a safe and effective modality for endoscopic treatment of RRP in adults and children when used appropriately [43,44,45]. Compared with lasers, powered instruments avoid the risk of thermal injury and airway fire [46]. Various blade types and sizes are available. The Skimmer blade (Medtronic Xomed) can be used for more precise removal of papillomas near the vocal fold, while the Tricut blade (Medtronic Xomed) can be used to remove more bulky lesions. Speed settings vary from 600 to 1400 RPM, with slower speeds used for the rapid removal of large lesions and faster speeds for more precision near normal mucosa [44]. Hemostasis is often achieved with the application of epinephrine-soaked pledgets [43,44].

Radiofrequency coblation (coblator) has become increasingly utilized in a variety of otolaryngological procedures [47,48,49]. It has been used safely and effectively in both adult and pediatric patients with extensive laryngeal papillomatosis [50,51]. The benefits of radiofrequency coblation have been reported by some to include reduced blood loss, minimal damage to underlying tissues, decreased operation time, and a greater time interval between additional procedures compared with laser ablation [52]. Ultimately, the choice of instrument is guided by surgeon preference, as none has been shown to be superior [37].

Regardless of the method, when addressing papillomas that involve the vocal fold, care must be taken to remain superficial to the vocal ligament [25]. A subepithelial injection of saline, epinephrine, or lidocaine into the superficial lamina propria can be utilized to protect the vocal ligament from damage. There is a significant risk of anterior web formation when treating the bilateral sides of the anterior commissure, and this can be staged to avoid this complication [25].

Figure 1 demonstrates a patient who had too aggressive anterior commissure laser treatment. A consequence of this was a very large anterior glottic web. Should this be encountered, the web may be lysed in the midline and steroid-injected (Figure 2). If the web is not the full thickness of the vocal folds, it should be cut with cold steel to help prevent the return of the web during healing. Once healed the web should be significantly improved (Figure 3). Consideration should be given to the dilation of the glottis following lysis.

Although less common, care must also be taken not to disrupt the mucosa along the entirety of the posterior commissure. Should this occur, a patient is at great risk of developing posterior glottic stenosis (Figure 4). This should also be treated similarly to the anterior glottic web with midline lysis of the scar and injection of steroids, and dilation should always be performed in these cases (Figure 5). The combination of anterior or posterior glottic stenosis in the setting of a patient with papillomas can lead to significant airway obstruction and should be avoided in all cases with careful surgical attention (Figure 6).

Whenever there is laryngeal involvement, it is critical to evaluate the trachea and mainstem bronchi to evaluate for distal spread of disease. Jet ventilation should be avoided if possible to prevent theoretical distal seeding of the disease. A tracheostomy should also be avoided unless necessary to prevent papillomatous involvement of the stoma.

Management of LP has traditionally required surgical excision within the operating room. However, in more recent years, office-based procedures have provided an attractive alternative, with the goal of saving costs and time and avoiding the risks of general anesthesia (Figure 7 and Figure 8). Office-based laser therapy has been reported as a safe and effective treatment that is well tolerated in unsedated patients [53,54,55].

Office-based laser treatment of LP is less costly than in the operating room [56,57]; however, these reports also found that patients required more frequent office-based treatments, which may reduce its cost-effectiveness [56,58]. Not all patients are suitable candidates for office-based procedures, including those with severe disease burden or airway compromise. Appropriate patient selection based on expected tolerance is also an important consideration. Management in the operating room while under general anesthesia offers a more controlled environment and access to more specialized instruments. There is now more of a trend to treat patients in the OR when they originally present, with further treatment offered in the office to delay, sometimes indefinitely, a repeat operation.

A concern that applies to both in-office and operative management of papilloma is the growing concern for the risk of transmitting HPV to providers during procedures in which viral particles may be vaporized because of electrocautery or laser ablation of HPV-infected tissues [59]. There is increasing evidence that ablative procedures disperse HPV DNA particles into the environment [60,61], while the infectivity of HPV DNA particles in ablative plumes is unclear. The infectivity of laser plumes containing viral particles has been demonstrated with bovine papillomavirus (BPV) in animal models [62,63]. Some data suggest increased rates of HPV DNA in samples taken from providers performing laser treatment of warts, yet the rates remain comparable to those of the general population [59,64]. In a series of 12 patients with laryngeal papillomatosis treated with KTP laser, HPV was undetectable by PCR on the laser fibers used to treat them [65]. The available evidence demonstrates that HPV DNA can be dispersed during ablative procedures using laser and electrocautery, with uncertain infectious potential. Providers are, thus, at theoretically greater risk of HPV exposure, and best practice includes utilizing N95 masks and smoke evacuation systems to minimize this risk [60].

## 5. Medical Management

While the primary management of LP is surgical excision, adjuvant therapy is utilized to control the disease in up to 20% of patients [66]. Table 1 summarizes treatment options. There are no indications for when to use adjuvant therapies, but it is typically considered in cases where more frequent operative treatment is needed or if there is extralaryngeal involvement [67]. Cidofovir is an antiviral agent that inhibits viral replication. It is a nucleotide analog that is incorporated into viral DNA and inhibits viral DNA polymerases [68]. It is FDA-approved for the treatment of CMV retinitis in patients with AIDS but has been used off-label as an intravenous and intra-lesional adjuvant treatment for RRP since the 1990s. Initial studies evaluating its efficacy in treating RRP showed significant rates of disease remission with intralesional injections [69,70]. Cidofovir has also been documented to reduce viral load and modified Derkay scores [71]. Many studies published since its initial use in treating RRP have demonstrated positive results with minimal side effects [67]. However, a randomized, double-blind, placebo-controlled study showed that there was no statistically significant difference in disease severity between those who received cidofovir compared with placebo [72]. A Cochrane review concluded that there is insufficient high-quality evidence to support the efficacy of intra-lesional cidofovir in the treatment of RRP [73].

Despite the widespread use of cidofovir in treating RRP, concerns regarding its risk of nephrotoxicity, neutropenia, and possible oncogenicity were raised. In animal studies, cidofovir was found to be carcinogenic and teratogenic [74]. Current reports do not show evidence of long-term nephrotoxicity, neutropenia, or the development of dysplasia or malignancy related to the use of intralesional cidofovir [67,75]. Bevacizumab is a humanized monoclonal antibody that inhibits angiogenesis by blocking vascular endothelial growth factor A (VEGF-A) [76]. VEGF-A mRNA is highly expressed in RRP tissue samples compared with unaffected tissue, implicating its role in the disease process [77]. Initial case reports showed dramatic responses with systemic therapy in both adult and pediatric patients [78,79,80]. More recent systematic reviews suggest that intralesional and systemic bevacizumab is effective in decreasing disease burden and the need for frequent surgical intervention in adults and children [76,81,82]. Systemic administration is particularly well-suited for lesions that involve the tracheobronchial tree and are more difficult to access for surgical excision [67]. While bevacizumab has been shown to be well tolerated with minimal adverse effects, boxed warnings suggest an increased risk of gastrointestinal perforation, impaired wound healing, and hemorrhage [81,83].

Intralesional bevacizumab is frequently used within the larynx at a concentration of 25 mg/mL. Although it has previously been reported that there is a maximum FDA dose of 50 mg/kg [84], this drug is currently not FDA-approved and is used off-label for papillomas. A prior review showed results from studies that had total dosages of 1.25 mg to 100 mg [81]. While many studies show improvements in disease severity and voice outcomes, it is important to note that there are limitations when comparing results from these studies as treatment protocols, dosages, intervals, and outcomes varied widely. Furthermore, intralesional injections are utilized as an adjuvant to surgical intervention and have not been evaluated without the confounding effects of primary surgical treatment.

A consensus statement on the administration of systemic bevacizumab for RRP published in December 2024 provided a treatment algorithm regarding initial dosing, intervals, tapering, monitoring, and reintensification of therapy [85]. Prior to initiating treatment, patients should undergo chest CT, renal function tests, blood pressure checks, and pregnancy testing where applicable. The initial dose is 10 mg/kg IV, spaced at 3–4-week intervals. Disease burden should be assessed via laryngoscopy/bronchoscopy and patient-reported symptoms every 3–4 cycles. Chest CT should be repeated every 1–2 years.

These guidelines recommended that if no response is observed after 3–4 cycles, systemic administration should continue for another 3–4 cycles, followed by reassessment. In patients who respond well to treatment after 3–4 cycles, the dose can be tapered by increasing the time interval by 3–4 weeks in between each cycle, using periodic laryngoscopy/bronchoscopy and patient-reported symptoms to assess for the maintenance of disease response. The treatment duration is indefinite, but ultimately, bevacizumab should be administered at the lowest possible frequency needed to maintain control of the disease. Their review showed the mean time to recurrence after cessation of treatment was 5.4 months, with a rapid response to treatment when therapy resumed [85]. Repeated surgical procedures can have a significant impact on quality of life and expose patients to the risks of anesthesia and laryngeal surgery. Systemic bevacizumab offers an opportunity to potentially avoid repeated surgeries.

**Table 1 cancers-17-00929-t001:** Treatment options.

Treatment Modality	Mechanism	Advantages	Limitations	Evidence	References
Surgical debulking	Removal of lesions via microdebrider, laser, and coblator	Mainstay of managing LP and airway obstruction	Frequent procedures, risks of general anesthesia, and cost	N/A	N/A
Office-based treatment	Laser ablation of lesions via transnasal flexible laryngoscopy	Disease maintenance, avoid surgery and general anesthesia, and reduced cost	Ability to tolerate office-based laser, frequent procedures, and not suitable for severe disease or airway compromise	Average time interval procedures: 10.59 months (OR) vs. 5.4 months (office); cost: USD 10,105 (OR) vs. USD 2081 (office)	Chen et al., 2021 [58]
Bevacizumab	Monoclonal antibody blocking VEGF-A; intralesional and systemic administration	Reduced frequency of surgical intervention; may address lesions on tracheobronchial tree	Frequent infusions every 3–4 weeks	Prolonged surgical interval in 95% of patients (systemic) vs. 62% (intralesional)	Pogoda et al., 2022 [82]
Cidofovir	Inhibits viral DNA polymerase; intralesional injection	Decreased disease recurrence and severity	Limited high-quality evidence to support its efficacy	No difference between cidofovir and placebo for disease severity, VHI, HRQoL, or number of procedures	Chadha et al., 2012 [73]
HPV vaccination	Antibody- and cell-mediated immunity against HPV 6, 11, 16, 18 (quadrivalent), 31, 33, 45, 52, and 58 (9-valent)	Prevents HPV infection and HPV-mediated disease. As treatment: reduced frequency of surgical intervention	Limited evidence of impact of national vaccination programs on rates of RRP; not approved for patients > 45 years	Mean ISI increased by 15.73 months after vaccination; mean number of procedures decreased by 4.43 per year after vaccination	Ponduri et al., 2023 [86]
Immune checkpoint inhibitors	Monoclonal antibody targeting PD-1 (Nivolumab, Pembrolizumab), PD-L1 (Avelumab)	Potential for future non-surgical treatment options	Limited evidence with few patients and mixed responses to treatment; ongoing clinical trials	Pembrolizumab: partial response in 57% of patients; stable disease in 33% Nivolumab: reduction in disease burden in 2/2 patients Avelumab: improvement in disease burden in all patients; partial response in 56% of patients; no improvement in pulmonary disease	Creelan et al., 2019 [87]; Allen et al., 2019 [88]; Pai et al., 2022 [89]

VHI: Voice Handicap Index; HRQoL: health-related quality of life; ISI: intersurgical interval.

## 6. HPV Vaccine

The greatest advancement in the medical management of this disease has not been treatment but, rather, prevention. In 2006, the quadrivalent HPV vaccination was first made available to the public, targeting HPV types 6, 11, 16, and 18. In 2015, a nine-valent vaccine was released, targeting additional types 31, 33, 45, 52, and 58. The HPV vaccine has also been proposed as potentially having a therapeutic effect in the treatment of RRP [90]. Theories suggest that both antibody- and cell-mediated immune responses may decrease the risk of recurrence and reinfection by inhibiting latent HPV at the surgical site [91,92]. A 2019 meta-analysis found that the use of the HPV vaccine as an adjuvant treatment for RRP reduced the number of surgical procedures [93]. A more recent systematic review in 2024 included new datasets that have since been published and also included two unpublished datasets, totaling 243 patients in the pooled sample [86]. The meta-analysis revealed that after vaccination with the quadrivalent or nine-valent HPV vaccine, the mean number of surgical procedures per year decreased by 4.43, and the mean intersurgical interval (ISI) increased by 15.73 months.

Vaccination against HPV is recommended for patients of all ages with RRP. However, it is currently only recommended by the CDC for patients aged 9 to 26 years, with consideration given up to 45 years based on discussion with the treating physician [94]. As a result, patients over 26 may encounter out-of-pocket costs. A study evaluating the efficacy of HPV vaccination in patients over the age of 45 revealed an increase in the ISI and a decrease in the number of procedures, consistent with previously published data [95]. This can aid clinicians when counseling patients outside the recommended age range on the efficacy of HPV vaccination, given its out-of-pocket cost.

The impact of national HPV vaccination programs on the incidence of RRP was investigated in a systematic review published in 2024, which showed a decrease in the incidence of J-RRP [96]. It included four studies from the United States, New Zealand, and Australia. National HPV programs were introduced in the United States in 2006, Australia in 2007, and New Zealand in 2008 [96]. In the United States, the incidence decreased from 2.0 per 100,000 in 2004–2005 to 0.5 in 2012–2013; in Australia, the incidence decreased from 0.16 per 100,000 in 2012 to 0.022 per 100,000 in 2016; and no significant decline in incidence was found in New Zealand [96]. The conclusions were limited by the small sample sizes (15–30 cases) and retrospective designs in the Australian and New Zealand studies. The United States study was a prospective cross-sectional design consisting of 576 cases across 26 tertiary medical centers in 23 states. This systematic review was the first to evaluate the incidence of J-RRP since the implementation of HPV vaccination programs and demonstrated an overall decline. It also indicated the need for more population-level data to better assess the impact that HPV vaccination programs have had on rates of RRP.

## 7. Future Directions

The work on improving and creating new vaccines continues and an ongoing phase 1/2 trial is evaluating DNA plasmid vaccines targeting the E6 and E7 proteins of HPV 6 and 11 [97]. Interim results have shown a decrease in the number of surgical interventions required compared with the year prior to administration, decreased severity, and a durable cellular response against HPV 6 and 11 [98]. While 5% of the population shows evidence of laryngeal HPV infection, only a very small portion of these patients develop RRP [99]. These patients typically have normally functioning immune systems and mount antibody responses to viral proteins, so it remains unclear why only some individuals develop papillomas [99]. Interest has focused on the role of immune pathways. The programmed death 1 (PD-1) and ligand (PD-L1) pathway facilitates tumor evasion of the immune system via T-cell suppression [99,100]. PD-1 is a T-cell receptor that suppresses T-cell function when activated by PD-L1 expressed on target cells [99]. PD-L1 expression has been observed in LP specimens along with infiltration of PD-1 T-lymphocytes [99,101]. The strongest expression of PD-L1 was observed in the basal papilloma layer; however, not all specimens expressed PD-L1 [99,101].

Nivolumab and pembrolizumab are monoclonal antibodies that block PD-1 and have been used to treat advanced head and neck cancer [100,102]. Few studies have evaluated the clinical utility of PD-1/PD-L1 blockade in the treatment of LP. In one study from 2019, two adult patients with recurrent J-RRP were treated with nivolumab [87]. One patient experienced remission after 9 months of treatment; the second patient experienced an improvement in laryngeal disease but had a mixed response with pulmonary lesions, where some shrank, and others grew [87]. A phase II trial evaluated avelumab (NCT02859454), a monoclonal antibody targeting PD-L1 [88]. The results showed improvement in laryngeal disease across all 12 patients, but pulmonary disease did not respond [88]. Another phase II study of pembrolizumab for RRP (NCT02632344) included 21 patients with either A-RRP or J-RRP [89]. In total, 57% of patients experienced a partial response, and the mean number of procedures decreased by seven surgeries per year [89]. These preliminary findings suggest a potential for the use of checkpoint inhibitors in the management of LP but are currently not approved for use in this setting.

HIV protease inhibitors have demonstrated activity against HPV-mediated cervical carcinoma in vitro, with lopinavir being the most effective one identified [103]. A study of patients with cervical HPV-positive high-grade squamous intraepithelial lesions (HSILs) treated with topical lopinavir/ritonavir showed no dysplasia in 63% of patients after 12 weeks of treatment [104]. RNA interference targeting E7 has also shown activity against cervical carcinoma in vitro via upregulation of p53 and retinoblastoma (Rb) protein, providing potential targeted gene therapy for HPV-mediated disease [105]. These potential therapies have demonstrated activity against HPV-mediated disease and may have the potential to treat RRP in the future.

One of the main challenges in evaluating the various treatment options for LP is the lack of prospective placebo-controlled trials with sufficient sample sizes. Treatment protocols and outcome measures vary widely, making it difficult to appropriately compare results across different studies. Further investigation carried out through multi-institutional studies with standardized protocols and outcome measures would aid in evaluating the various treatment options with larger samples.

## 8. Conclusions

RRP is the most common benign disease of the larynx, caused by HPV. It is notoriously challenging to treat due to its tendency to recur. Two distinct forms are recognized based on the age of diagnosis, with J-RRP representing a more aggressive disease. The mainstay of management is surgical excision with many different modalities utilized according to surgeon preference. Patients typically require frequent surgical procedures, and office-based laser treatment is useful in delaying repeat trips to the operating room in appropriately selected patients. Many adjuvant treatments are available with mixed responses but may be particularly suited for patients with distal disease that is not easily accessible for resection. Ongoing trials are exploring new vaccine strategies and targeted therapies that have the potential to improve control of LP in the future.

## Figures and Tables

**Figure 1 cancers-17-00929-f001:**
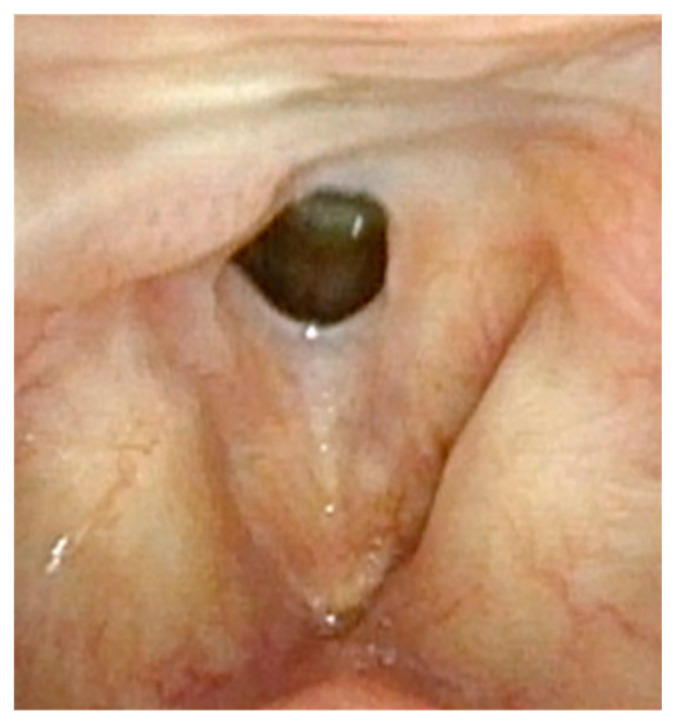
Large anterior glottic web with small amount of papilloma on right posterior commissure.

**Figure 2 cancers-17-00929-f002:**
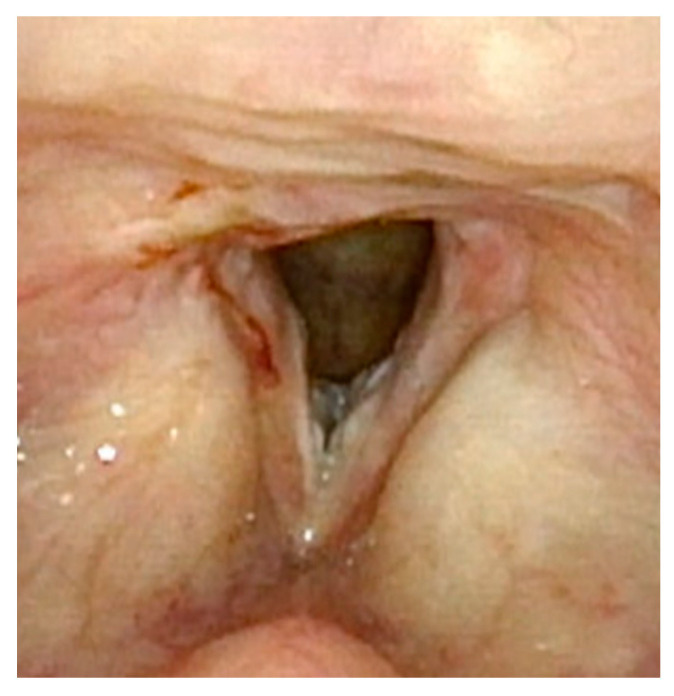
One week post-op following laser ablation of laryngeal papilloma and lysis of anterior glottic web.

**Figure 3 cancers-17-00929-f003:**
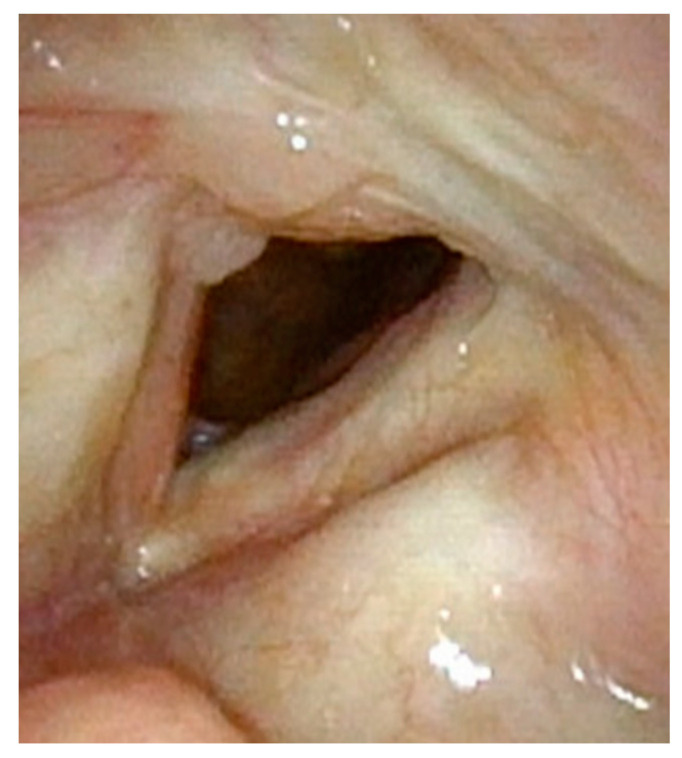
Three months post-op following laser ablation of laryngeal papilloma and lysis of anterior glottic web. The majority of the anterior glottic web is resolved, but there is already regrowth of papillomas.

**Figure 4 cancers-17-00929-f004:**
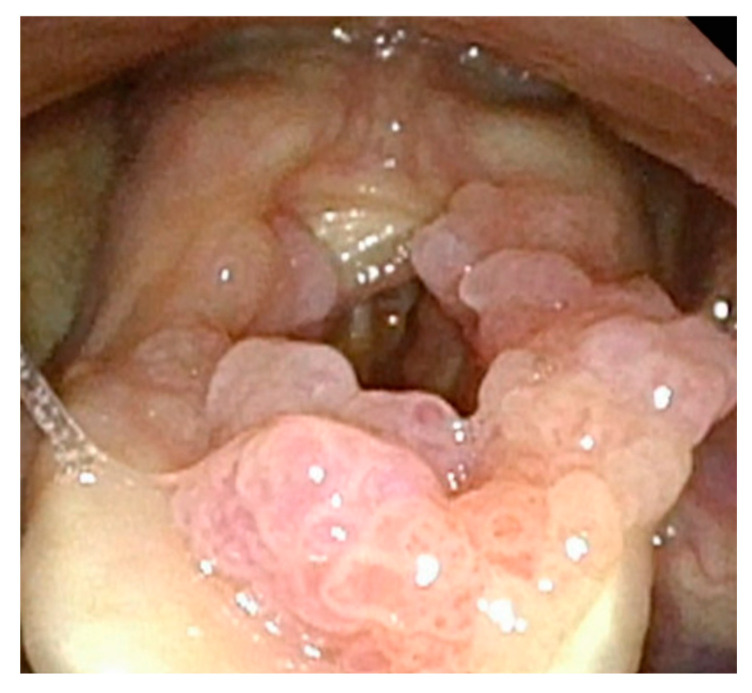
Significant papillomatous growths throughout supraglottis and severe posterior glottic web.

**Figure 5 cancers-17-00929-f005:**
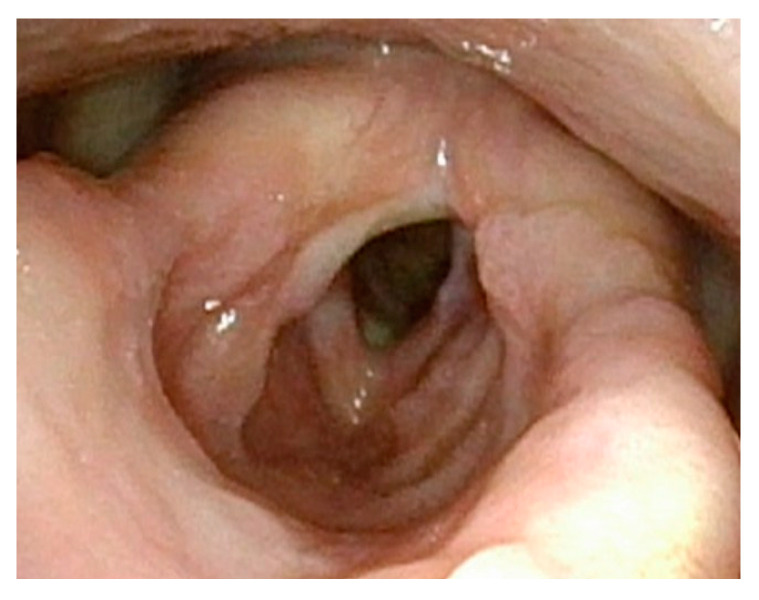
Two months post-op after lysis of posterior glottic web and ablation of papillomas. Web has significantly improved, and there is already regrowth of papillomas.

**Figure 6 cancers-17-00929-f006:**
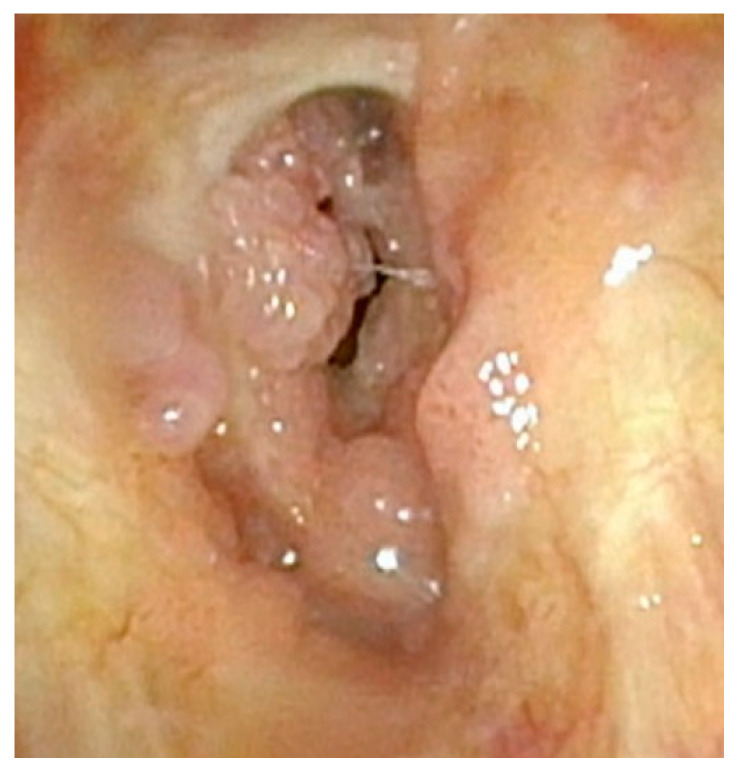
Significant posterior glottic web with papillomas causing near-complete obstruction of the airway.

**Figure 7 cancers-17-00929-f007:**
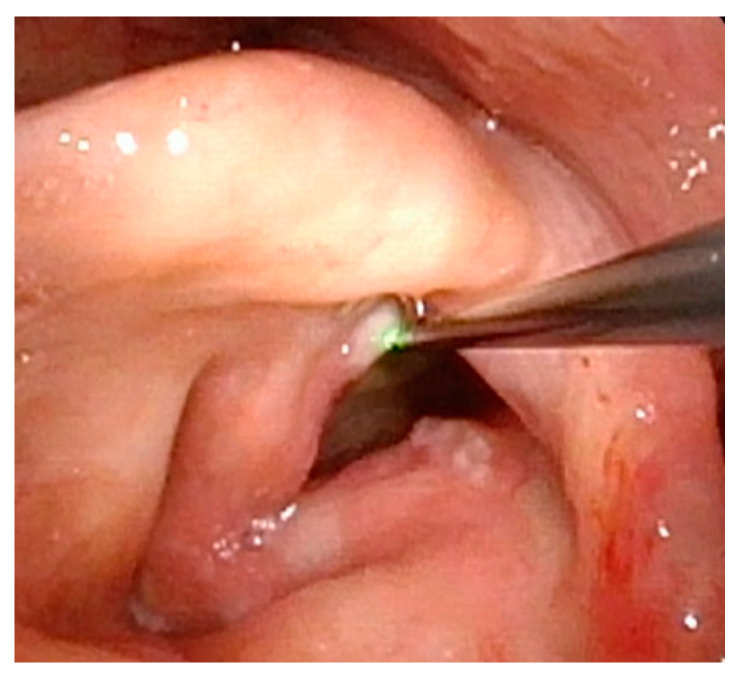
Scattered papillomas through glottis with true blue laser beginning to ablate papilloma on the posterior right vocal fold.

**Figure 8 cancers-17-00929-f008:**
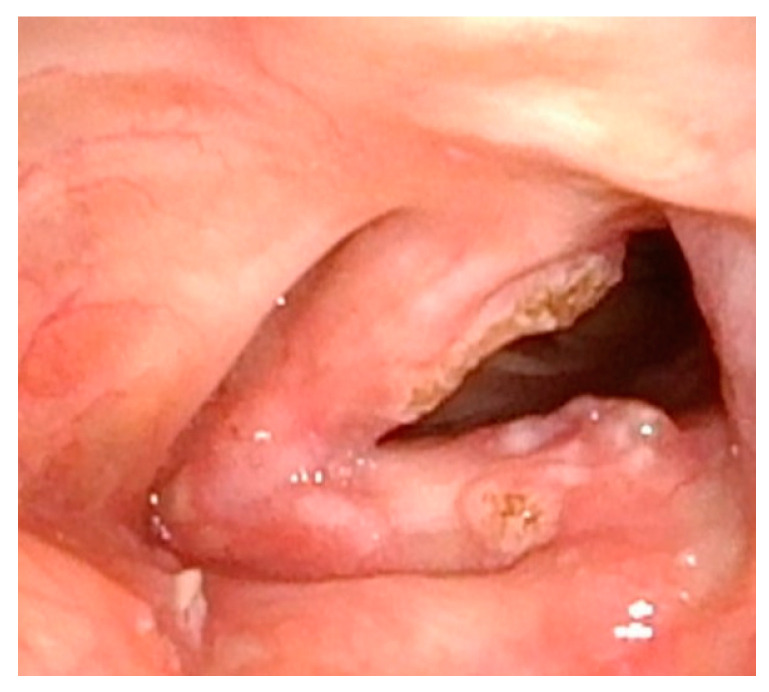
Completion of in-office laser treatment. Papillomas are either fully ablated or blanched white with the expectation of necrosis over the following week.

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
