# Peer review of "Laryngeal Papillomatosis"

_cancers, 2025, doi:10.3390/cancers17060929_

Round 1

Reviewer 1 Report

Comments and Suggestions for Authors

 This study provides a comprehensive review of recurrent respiratory papillomatosis (RRP), a benign yet challenging disease caused by HPV types 6 and 11. It discusses the clinical presentation, bimodal onset (juvenile and adult), and the disease's recurrent nature, often requiring frequent surgical interventions. The management focuses on various surgical techniques such as COâ‚‚ lasers, KTP lasers, microdebriders, and coblation, alongside adjuvant therapies like cidofovir and bevacizumab, though their efficacy is variable. I would like to offer the following points for consideration by the authors towards the improvement of the manuscript:

1-Abrupt shifts between sections make the manuscript difficult to follow. For instance, moving from HPV pathophysiology directly into the clinical presentation of RRP without contextualizing the connection feels disjointed.

1- The section on bevacizumab in your manuscript requires significant enhancement to provide clarity and utility for readers. Specifically, the discussion lacks details on dosing, frequency, and administration, which are critical for clinicians. While you mention its potential efficacy, the information provided is vague and unsupported by detailed data from the existing literature.

2- This manuscript lacks any tables or figures, which significantly limits its accessibility and utility for readers. High-quality review articles typically include visual aids to summarize complex information, enhance readability, and engage the audience.

3-Many cited studies predate 2020, with critical updates in HPV vaccines, immunotherapy, and surgical innovations overlooked

4- The potential of immunotherapies (e.g., PD-1 inhibitors) is briefly mentioned but lacks detailed discussion of their mechanisms, challenges, or implications for clinical use in RRP.

5-Combine related sections (e.g., surgical modalities) into a cohesive discussion with comparative analyses.

6- Strengthen the "Future Directions" section by highlighting emerging trends and research gaps.

Author Response

1-Abrupt shifts between sections make the manuscript difficult to follow. For instance, moving from HPV pathophysiology directly into the clinical presentation of RRP without contextualizing the connection feels disjointed.

Response 1: Several changes have been made to the manuscript to help with the flow.  We believe it should no longer feel disjointed.  The number of individual sections has been greatly decreased with several being combined together to help with this as well.

1- The section on bevacizumab in your manuscript requires significant enhancement to provide clarity and utility for readers. Specifically, the discussion lacks details on dosing, frequency, and administration, which are critical for clinicians. While you mention its potential efficacy, the information provided is vague and unsupported by detailed data from the existing literature.

Response 1: The entirety of this section has ben changed.  It has also been moved into the "Medical Management" section.

2- This manuscript lacks any tables or figures, which significantly limits its accessibility and utility for readers. High-quality review articles typically include visual aids to summarize complex information, enhance readability, and engage the audience.

Response 2: Eight figures have been added to this manuscript to help illustrate several points and encourage engagement. 

3-Many cited studies predate 2020, with critical updates in HPV vaccines, immunotherapy, and surgical innovations overlooked

Response 3: We appreciate this suggestion and several newer studies have been added to our citations including many from 2024. 

4- The potential of immunotherapies (e.g., PD-1 inhibitors) is briefly mentioned but lacks detailed discussion of their mechanisms, challenges, or implications for clinical use in RRP.

Response 4: We have greatly expanded our discussion on PD-1 inhibitors and can be found in the "Future Directions" section.

5-Combine related sections (e.g., surgical modalities) into a cohesive discussion with comparative analyses.

Response 5: These and other sections have been combined as recommended.

6- Strengthen the "Future Directions" section by highlighting emerging trends and research gaps.

Response 6: We have significantly expanded on the "Future Directions" sections as recommended.

Reviewer 2 Report

Comments and Suggestions for Authors

The submitted manuscript is not an original research article, nor is it a review or systematic review of the literature. The authors have comprehensively collected the current knowledge on laryngeal papillomatosis from its etiology, through the methods of treatment used, to future perspectives. The article has an undoubted educational value, presents the issue in a concise way and is based on well-selected literature. If the authors intend to write a short educational article, then it meets such criteria. However, if it were to be a scientific overview of the subject, then it lacks a broad, traditional discussion of individual topics.

Author Response

Thank you for your comments.  We have significantly added to each out these topics to give it more depth.

Round 2

Reviewer 1 Report

Comments and Suggestions for Authors

I am satisfied that the authors have addressed all of my previous concerns about the article. It is now much improved and I feel that it is now suitable for publication.

Author Response

Response: Table has been added as requested.